# Phosphorus Recycling, Biocontrol, and Growth Promotion Capabilities of Soil Bacterial Isolates from Mexican Oak Forests: An Alternative to Reduce the Use of Agrochemicals in Maize Cultivation

Rocío Hernández-León [1], Antonio González-Rodríguez [1] and Yunuen Tapia-Torres [2,*]

[1] Instituto de Investigaciones en Ecosistemas y Sustentabilidad, Universidad Nacional Autónoma de México, Morelia 58190, Mexico

[2] Escuela Nacional de Estudios Superiores unidad Morelia, Universidad Nacional Autónoma de México, Morelia 58190, Mexico

* Correspondence: ytapia@enesmorelia.unam.mx

**Abstract:** Six bacteria (*Bacillus velezensis* 13, *Bacillus subtillis* 42, *Pseudomonas fluorescens* E221, *Pseudomonas Poae* EE12, *Rahnella* sp. EM1, and *Serratia* sp. EM2) isolated from the soil and litter of Mexican oak forests were characterized by identifying their ability to acquire phosphorus from different sources, analyzed for their biocontrol capabilities against two different phytopathogenic fungi, and finally tested for their ability to stimulate the germination of maize seeds and promotion of maize seedling growth. The greatest capacity to biocontrol the mycelial growth of phytopathogenic fungi *Botrytis cinerea* and *Fusarium oxysporum* was found in *B. velezensis* 13 and *B. subtillis* 42. *P. poae* EE12 and *P. fluorescens* E221 significantly promoted germination and the length of the primary root in *Zea mays*. *Rahnella* sp. EM1 and *Serratia* sp. EM2 could produce indole compounds related to auxin synthesis and increased the fresh weight of the maize seedlings. Together, these isolates represent an alternative to reduce the use of agrochemicals in maize cultivation. In general, soil microorganisms from Mexican oak forests represent a source of genetic resources for the sustainable management and conservation of soils for agricultural use.

**Keywords:** phosphorus mineralization; solubilization; PGPB; oak forest

## 1. Introduction

The use of fertilizers to increase agricultural production and pesticides to avoid losses to pathogenic organisms has been a solution against human starvation, despite having negative impacts on the ecosystems, such as the reduction of the diversity of soil microorganisms and the increase of water contamination [1,2]. Nitrogen (N)- and phosphorus (P)-based fertilizers are used in the largest quantities worldwide [3–5]. However, the production of P fertilizers currently represents a major challenge since it is not possible to synthesize P. Additionally, in most natural ecosystems, P is a major limiting factor of primary productivity [6], due to its importance for cellular functioning and biomass increase [7] and its relatively low availability for the biota. Therefore, agricultural production is highly dependent on the addition of large quantities of P in the form of different molecules that are easily available for the plants (water soluble phosphate fertilizers; WSPF). The P in these fertilizers is derived from mines of sedimentary deposits of ancient phosphorites [8]. Thus, phosphate rock, used intensively for food production, is a fossil resource that is rapidly running out due to its unsustainable use, mainly in the agricultural sector [9,10].

In ecosystems, P can be found in both inorganic and organic forms [11–13]. However, inorganic phosphate (Pi; $HPO_4^{2-}$, and $H_2PO_4^{-}$) dissolved in soil solution is the main P source for plants and microorganisms, but its availability in soil is very low due to its high reactivity [14]. When the supply of Pi is not enough to satisfy the biological

demand, Pi acquisition is carried out through geochemical and biological recycling of different organic and inorganic chemical P forms; also known as P fractions, which can be found in soil [15]. The concentration of each fraction is determined by the processes of mineralization of organic P (Po) and immobilization of Pi, as well as by the balance between adsorption–desorption and precipitation–solubilization of Pi [12].

The solubilization and mineralization processes are the basis of the P biological recycling and of the increase of soil Pi availability. Both processes occur extracellularly: mineralization is a microbial enzyme-catalyzed process and solubilization that occurs as a sub-product of an intracellular enzyme-catalyzed process [16]. Different enzymes are necessary for the mineralization of organic P compounds (phosphate esters, phosphonates, and phytates), such as phosphatases, C-P lyases, phosphonates, and phytases, and their synthesis in the soil depends on the presence of microorganisms with the genetic material to express the necessary genes.

In addition to the challenge facing the agricultural sector of increasing the availability of P in the soil, large areas of crops are lost annually worldwide due to pests [17–19]. In Mexico, 78.2% of agricultural production units report losses due to climatic causes, pests, and diseases [20,21]. Among phytopathogens that cause many agricultural losses, the genus *Fusarium* includes plant pathogenic fungi with a wide variety of hosts and infection strategies [22–24]. One of the most relevant species is *Fusarium oxysporum*, which can invade roots and cause wilt diseases through colonization of the xylem tissue [25,26]. The pathogen *Botrytis cinerea* also causes serious losses in more than 200 species of crops worldwide. It is very destructive and difficult to control because it has a variety of attack modes and diverse hosts, including both dicotyledonous and monocotyledonous species [27].

Fortunately, there are alternatives to increase the production and decrease the concentrations of agrochemicals currently used. For this, it is necessary to implement a series of agroecological practices such as biofertilization and biological pest control [28,29]. Plant growth promoting bacteria (PGPB) constitute a key functional group that favors crop development through different processes such as the synthesis of phytohormones, which can act to enhance or regulate different stages of plant growth.

The phytohormones produced by some bacteria can increase the area of the roots, helping plants to have a greater absorption of water and other nutrients (N and P) from the soil as well as a greater height and grain yield [30]. Auxins are quantitatively the most abundant phytohormones secreted by most plant-associated bacteria [31], which influence processes such as phototropism and gravitropism, the formation of floral organs, leaves, lateral roots, the maintenance of the individuality of the apical meristem of the shoot and the root in addition to playing a fundamental role in the early embryo pattern [32–34]. Local auxin biosynthesis is also involved in the early stages of pollen development and shade avoidance syndrome in plants as well as root growth in response to aluminum stress [35,36]. The main natural auxin is indole-3-acetic acid (IAA), and its main precursor is tryptophan. From it, at least five different pathways have been described for the synthesis of IAA in bacteria and most of the pathways show similarities with the ones described in plants [37]. In several microorganisms, the redundancy of IAA biosynthetic pathways has been observed, meaning that multiple pathways are present and active in a single microorganism. Some of the intermediates from which the different synthesis pathways take their name are 3-indoleacetic and indolepyruvic acids [36–38].

Different groups of bacteria also possess the ability to control soil-borne plant pathogenic fungi. The *Pseudomonas* genus has been reported as a producer of a large group of metabolites such as 2,4-diacetylphloroglucinol, phenazine-1-carboxylic acid, and pyrrolnitrin, active against a wide spectrum of phytopathogenic fungi. Oligomycin A, kanosamine, zwittermicin A, and xanthobaccin produced by *Bacillus*, *Streptomyces*, and *Stenotrophomonas* spp. are compounds identified as antibiotics [39]. In addition to these compounds, *Rahnella aquatilis* can produce siderophores that could have significant antimicrobial activities in vitro and that can help regulate the microflora of plants [40].

These PGPB can live anywhere. In Mexico, oak forests harbor a great biological diversity and therefore a great functional diversity [41] that have been little analyzed as a source of potentially useful microorganisms. Despite being few, there are studies that indicate that isolated bacteria from oak forests have different capacities to promote plant growth that, if used in the correct way, can help to increase the yield of different crops sustainably [42–44].

This work offers an approach to select soil bacteria that constitute an alternative to the treatment of crops with fertilizers and pesticides. In addition, the understanding of the stimulus-response mechanism between PGPB and plants represent an important contribution to propose a sustainable and adequate management of agricultural soils. Therefore, the main objective of this work is to evaluate the phosphorus recycling, biocontrol, and growth promotion capabilities in maize, of six bacteria isolated from Mexican oak soil forest, to determine if they can be used as potential bioinoculants to reduce the use of agrochemicals used in these crops.

## 2. Materials and Methods

### 2.1. Biological Material

We randomly selected 100 strains from the microbial collection of the Microbiomics Laboratory, National School of Higher Studies (Escuela Nacional de Estudios Superiores [ENES]) Morelia, National Autonomous University of México (Universidad Nacional Autónoma de México [UNAM]). The bacterial strains that we selected were isolated by serial dilutions and were incubated in nutrient agar medium for 24 h at 28 °C and reseeded until obtaining pure cultures. The bacteria in the collection were obtained using an inoculum of soil or litter for each sample from each site. Samples were obtained from oak forests in Llanitos, Michoacán, México (17°57′ N, 102°12′ W) and Avándaro, Estado de México, México (19°06′ N, 100°07′ W). Fresh samples were added to an Eppendorf tube with 900 μL of modified universal buffer (MUB) containing per 500 mL: Tris-hydrochloric aminomethane (6.05 g), maleic acid (5.8 g), boric acid (3.15), and citric acid (7.0 g). The resulting suspension was mixed continuously for 60 min and then used as an inoculum for a subsequent dilution in an Eppendorf tube with 900 μL of MUB. Four dilutions of each sample were plated onto Petri dishes with nutrient agar medium for 24 h at 28 °C. Colonies with different morphotypes (i.e., size, shape, and color) were selected. Purification was performed by sub-culturing on the same medium to ensure that the culture was axenic. The bacteria were kept in nutrient agar at 4 °C for subsequent experiments and stored in glycerol at −80 °C for long-term preservation.

### 2.2. Siderophores Production and Phosphate Solubilization

Tests for siderophore production were carried out in triplicate, inoculating the bacteria to be evaluated by puncture in Chrome Azurol Sulfonate Agar (CAS) plates [45] and incubating for 24 h at 28 °C. The production of siderophores was evidenced by the formation of yellow or orange halos around the bacterial colonies.

To evaluate the ability of bacterial isolates to solubilize phosphates, Pikovskaya Agar was used as the culture medium. (PY g/L: Yeast extract, 0.5; Dextrose, 10; $Ca_3(PO_4)_2$, 5; $(NH_4)_2SO_4$, 0.5; KCl, 0.2; Mg $SO_4$ + 7$H_2O$, 0.1; $MnSO_4$ + 6$H_2O$, 0.0001; $FeSO_4$ + 6$H_2O$, 0.0001; Agar, 15; Bromocresol purple, (0.1)). Tests were performed in triplicate, inoculating the bacteria to be evaluated on the plate with PY agar by puncture and incubating for 24 h at 28 °C. The formation of yellow halos and clearing zones, surrounding the bacterial colony, indicates the ability of the bacteria to solubilize phosphate.

### 2.3. Antifungal Activity

The selected 100 bacterial strains were subjected to antagonism tests against *Botrytis cinerea* and *Fusarium oxysporum* to evaluate their ability to inhibit mycelium growth. The phytopathogenic fungi *Botrytis cinerea* and *Fusarium oxysporum* were provided by the Biological-Chemical Research Institute (Instituto de Investigaciones Químico Biológicas-Universidad Mi-

choacana de San Nicolás de Hidalgo (IIQB-UMSNH) and Agricultural and Forestry Research Institute -Universidad Michoacana de San Nicolás de Hidalgo (IIAF-UMSNH), respectively.

An interaction test was carried out in Petri dishes containing potato dextrose agar and nutrient agar in a 1:1 ratio. A 9 mm inoculum of the phytopathogen was sowed in the center of the petri dish and, around it, 20 bacterial isolates to be tested were inoculated. Petri dishes were incubated at 28 °C for 10 days. Bacterial isolates that inhibited fungal growth were selected. Tests were carried out in triplicate.

After this initial screening, six bacterial isolates were selected (isolates named as 13, 42, EE12, E221, EM1, and EM2). Each selected isolate was then tested individually with each of the phytopathogenic fungi to determine the inhibition percentage. For this, a Petri dish was divided into four quadrants, as described in Hernández-León et al. [46]. In each quadrant, an inoculum of the fungus was placed, while the bacteria were seeded by streaking two perpendicular lines from one end of the petri dish to the other. The petri dishes were incubated at 28 °C in darkness for 6 days, at the end of this period the growth of the mycelium was measured. The test was carried out in triplicate and with a control group without bacteria, also in triplicate. The inhibition growth percentage was calculated using the following formula:

$$\text{Inhibition (\%)} = \frac{\text{Control average diameter} - \text{Treatment diameter}}{\text{Control average diameter}} \times 100$$

### 2.4. Molecular Characterization of Bacterial Isolates

DNA from isolates was purified using the QiaGen DNeasy UltraClean Microbial Kit following the manufacturer's instructions and corroborated by 1.0% agarose gel electrophoresis. We amplified the 16S rRNA gene by PCR using previously the published primer sequences 27F (5′-AGAGTTTGATCMTGGCTCAG-3′) and 1492R (5′-TACGGYTACCTTGTACGACTT-3′) [47]. The total volume of the reaction mixture was 25 µL, which contained 1 µL of DNA, 10 µL of 2x QIAGEN Multiplex PCR Master Mix, 1 µL of each target specific primers (10 µM), and 12 µL RNase-free water. The PCR conditions used were 95 °C for 3 min, followed by 30 cycles of 95 °C for 45 s (denaturation), 63 °C for 30 s (annealing), and 72 °C for 1 min (elongation), with a final extension of 5 min at 72 °C. Amplification was corroborated by 1.5% agarose gel electrophoresis. The PCR products were sequenced in Macrogen Maryland, USA.

### 2.5. Growth Test in Different Phosphorus Sources

To evaluate the capability of the isolates to utilize different P substrates, we used a defined medium (DM) and five different P sources. Initially, isolates were inoculated in defined medium without phosphorus to ensure the depletion of P storage and then passed to DM with five different substrates utilized as the sole phosphorus source: potassium phosphate ($KH_2PO_4$), aluminum phosphate ($AlPO_4$), calcium phosphate ($Ca(H_2PO_4)2H_2O$), 2-aminoethylphosphonic acid or 2AEP ($H_2NCH_2CH_2P(O)(OH)_2$) (phosphonate), ferric phosphate ($FePO_4$), and DM without phosphorus (-P) as the negative control. The base medium (DM) contained, per liter: Tris base, 6.057 g adjusted to pH 8.0; $NH_4NO_3$, 0.26 g; $MgSO_4$, 0.48 g; disodium citrate, 1.99 g; $ZnCl_2$, 0.000136 g; NaCl, 5 g; $FeCl_3$, 0.27 g; KCl, 0.1 g; $MnCl_2$, 0.2 g; CaCl, 0.4 g; glucose, 9 g; and amino acid mixture, 0.93 g. Heat-labile substrates (vitamin B complex, biotin, and nicotinic acid) were filtered, sterilized, and added aseptically after autoclaving. The ability to grow with different P sources was evaluated and after each isolate was transferred to a second DM Petri dish alongside the same five P sources, along with a control plate lacking P. The Petri dishes were incubated at 28 °C for 72 h. The growth after the second transfer was taken as the ability of the bacterial isolates to use the source of phosphorus [13].

### 2.6. Indole Detection

We qualitatively determined if a bacterium could produce indole compounds such as 3-indoleacetic acid and indolepyruvic acid, which are involved in auxin biosynthesis,

since these compounds can impact the development of plant roots [31] or be a part of the defense response against leaf pathogens [48]. A colorimetric method was used based on the oxidation of the indole ring with the Salkowski reagent ($H_2SO_4$ 42% + $FeCl_3$ 3%) [49]. The six previously selected isolates grew for 72 h at 28 °C in nutrient broth supplemented with 0.1% tryptophan. After that, 3 mL of the culture was centrifuged, and then 2 mL of the supernatant was mixed with 4 mL of Salkowski's reagent and incubated for 30 min at 28 °C. The strains that produced a turn to pink are reported as positive (+), and those that did not produce any color change were reported as negative for this test (−).

### 2.7. Maize Seed Germination Test

To test if the analyzed strains can modify the germination patterns, we used local maize seeds. The maize seeds were donated by farmers in the Huetamo region in Michoacán state, México. We inoculated 200 seeds with each of the strains (isolates named 13, 42, EE12, E221, EM1, and EM2), 200 were used as control (without inoculation), and we evaluated their germination. Seeds were sterilized with 5% sodium hypochlorite and washed with sterile water four times to remove the sodium hypochlorite [50].

The seeds were immersed for 30 min in a 30 mL overnight liquid culture of each isolate in nutrient broth at a concentration of $1 \times 10^7$ colony forming units CFU/mL. Control seeds were immersed in nutrient broth. After inoculation, seeds were dried for 90 min in a paper towel in a sterile environment. Sixteen square Murashige and Skoog (MS) Agar Petri dishes (11.8 cm side) with 12 or 13 inoculated seeds each arranged equidistant for a total of 200 seeds per treatment were incubated at 24 °C for 9 days in a growth chamber monitored daily with 12/12-h light/dark cycle at 24 °C because this is the optimum temperature for maize development [50]. The number of germinated seeds per Petri dish was counted daily to determine the germination percentage. The germination percentage was calculated using the following formula:

$$\text{Germination (\%)} = \frac{\text{number of germinated seeds}}{\text{total experimental seeds}} \times 100$$

The root length of each germinated seed was measured on the 10th day with the help of ImageJ software [51].

### 2.8. Early Growth Promotion Test in Seedbeds

To find out if the strains promoted the growth of maize seedlings during the early stages, seeds were sterilized with 5% sodium hypochlorite and washed with sterile water four times to remove the sodium hypochlorite [50]. Twenty-four seeds were treated with each bacterial strain in liquid culture (0.5 mL/seed) for 30 min and let dry afterwards for 90 min in a paper towel in a sterile environment.

Inoculated maize seeds were sown one per pot with sterile pine bark substrate and placed in a growth chamber for a 12/12-h light/dark cycle at 24 °C. All seedbeds for each treatment were watered with 300 mL of sterile water three times a week. After that, 1 mL of liquid culture of each of the PGPB was centrifuged and washed with Modified Universal Buffer (MUB) pH 7 for a second inoculation on the eighth day. Plants were harvested on the 16th day; photographs were taken of each plant to determine the total area with the ImageJ software [52]. The shoots were detached from the root and weighed separately, oven-dried at 75 °C for 48 h, and then weighed again.

### 2.9. Statistical Analysis

Significance of differences in primary root length, area, root weights, and shoots weights were tested with one-way analyses of variance (ANOVA) and post-hoc Tukey-Kramer HSD test in the JMP program [53].

## 3. Results

### 3.1. Selection and Molecular Characterization of the Bacterial Isolates

Six bacterial isolates named 13, 42, EE12, E221, EM1, and EM2 were selected based on their biocontrol capacity (see below). On the basis of the 16S rDNA gene sequencing data, bacterial isolate 13 was classified as *Bacillus velezensis* with 100% identity, isolate 42 showed 100% identity with *Bacillus subtillis*, isolate E221 showed 99.7% identity with *Pseudomonas fluorescens*, isolate EE12 was identified as *Pseudomonas poae* with 100% identity, isolate EM1 showed 99.5% identity with *Rahnella* sp., and isolate EM2 showed 100% identity with *Serratia* sp. (NCBI accession numbers: OK324128, OK324129, OK324130, OK324131, OK324132 and OK324133)

### 3.2. Siderophores Production and Phosphate Solubilization

Siderophores production was evaluated in solid medium, and it was observed that *Pseudomonas fluorescens* E221 is the strain that produced the largest halo in the CAS medium, followed by *Serratia* sp. EM2 and *Bacillus subtillis* 42. The rest of the strains did not produce a halo around them, indicating that there is no production of molecules that can release iron from chromogen. In PY agar, all the strains produced yellow halos around them, which indicates that they can solubilize the calcium phosphate in the medium. However *Pseudomonas fluorescens* E221, *Serratia* sp. EM2, and *Bacillus velezensis* 13 were the ones that produced larger halos, followed by *Bacillus subtillis* 42 and the rest of the strains (Figure 1).

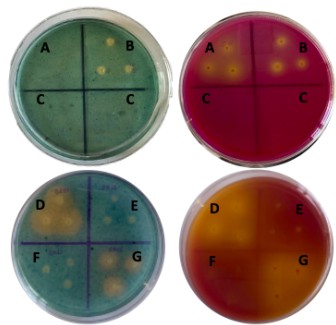

| Strains | Siderophores production (mm) | Phosphorus solubilization (mm) |
|---|---|---|
| *Bacillus velezensis* 13 | - | 13.6 |
| *Bacillus subtillis* 42 | 2 | 11.6 |
| *Pseudomonas fluorescens* E221 | 9.3 | 20.3 |
| *Pseudomonas poae* EE12 | - | 5 |
| *Rahnella* sp. EM1 | - | 4 |
| *Serratia* sp. EM2 | 7.3 | 15 |

**Figure 1.** Siderophores production and phosphate solubilization in solid medium (CAS (blue) and PY (red), respectively). Representative plates where the formation of halos is observed (left) where A refers to strain 13; B refers to strain 42; C corresponds to the agar control where no bacteria was grown; D refers to strain E221; E refers to strain EE12; F refers to strain EM1, and G refers to strain EM2 and average diameter of the halos in millimeters (right).

### 3.3. Biocontrol Capacity

Strains *B. subtillis* 42, *P. poae* EE12, and *B. velezensis* 13 reduced the size of the *F. oxysporum* mycelium by 17.3, 14.9, and 13.5%, respectively (Figure 2; Table 1). On the other hand, *Rahnella* sp. EM1, *B. subtillis* 42, and *B. velezensis* 13 decreased the growth of *B. cinerea*; however, the strains that showed a greater inhibition capacity were *B. subtillis* 42 and *B. velezensis* 13, with an inhibitory percentage of 46% and 92%, respectively. We can consider these isolates as potential biocontrol agents since *B. velezensis* 13 dramatically inhibited the growth of the fungus under our experimental conditions (Figure 2). Additionally, we evaluated the production of indolic compounds by the strains, as a desirable characteristic in plant growth promoting bacteria (Table 1).

### 3.4. Phosphorus Growth Test

Of the isolates analyzed, strains *B. velezensis* 13, *B. subtillis* 42, *P. fluorescens* E221, *Rahnella* sp. EM1, and *Serratia* sp. EM2 were able to use all sources of phosphorus tested (Table 2). *P. poae* EE12 was unable to use 2AEP as a source of phosphorus. All strains were able to use calcium phosphate as the only source of phosphorus, which corroborates the

data obtained in the analysis with PY medium. None of the strains were able to grow in media without phosphorus.

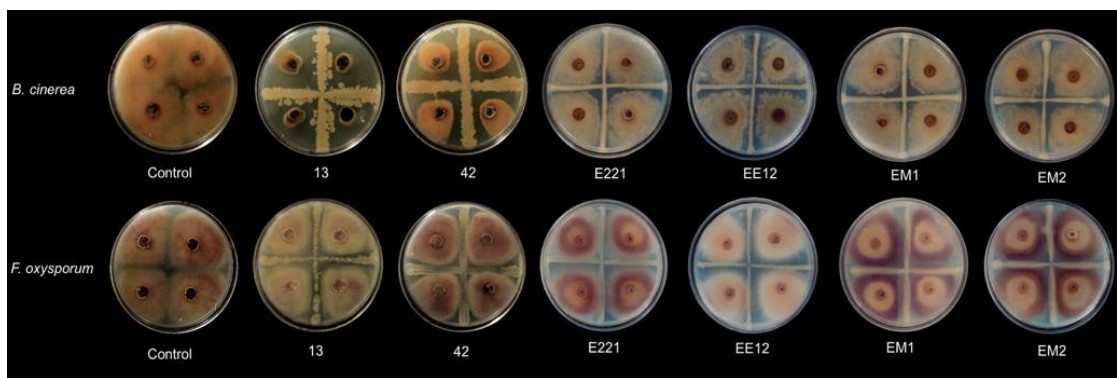

**Figure 2.** Effect of strains 13, 42, E221, EE12, EM1, and EM2 on the growth of the mycelium of *Botrytis cinerea* (upper row) and *Fusarium oxysporum* (lower row). The bacterial isolates were streaked onto Petri dishes in a cross shape, and mycelial plugs of 4 mm in diameter were deposited in the center of each of the quadrants. Control without bacteria.

**Table 1.** Identification, biocontrol analysis, and production of indole compounds by the six studied strains.

| Strains | Molecular Analysis | Biocontrol Analysis | | Biochemical Analysis |
|---|---|---|---|---|
| | Identification of Isolates Based on Partial Sequencing of the 16S rRNA Gene | Inhibition of *B. cinerea* Radial Growth (%) | Inhibition of *F. oxysporum* Radial Growth (%) | Indole Compounds Production |
| 13 | *Bacillus velezensis* | 92.4 | 13.5 | - |
| 42 | *Bacillus subtillis* | 46.9 | 17.3 | - |
| E221 | *Pseudomonas fluorescens* | 7.1 | 5.9 | + |
| EE12 | *Pseudomonas poae* | 8.9 | 14.9 | - |
| EM1 | *Rahnella* sp. | 12.1 | 6.1 | + |
| EM2 | *Serratia* sp. | 8.2 | 5.5 | + |

**Table 2.** Ability of strains to grow on substrates used as the sole phosphorus source.

| Strains | Phosphorus Sources Used by PGPBs | | | | | |
|---|---|---|---|---|---|---|
| | $AlPO_4$ | $Ca(H_2PO_4)2H_2O$ | 2AEP | $FePO_4$ | $KH_2PO_4$ | -P |
| *Bacillus velezensis* 13 | + | + | + | + | + | - |
| *Bacillus subtillis* 42 | + | + | + | + | + | - |
| *Pseudomonas fluorescens* E221 | + | + | + | + | + | - |
| *Pseudomonas poae* EE12 | + | + | - | + | + | - |
| *Rahnella* sp. EM1 | + | + | + | + | + | - |
| *Serratia* sp. EM2 | + | + | + | + | + | - |

The ability of the strains to metabolize different P sources in vitro indicates that their inoculation into seeds could improve the utilization of soil P through two different processes: solubilization of soil inorganic phosphorus compounds (secondary minerals) [52] as the inorganic molecules tested ($FePO_4$, $AlPO_4$, $KH_2PO_4$, and $Ca(H_2PO_4)_2H_2O$), and through the mineralization of organic molecules such as phosphonates (2AEP ($H_2NCH_2CH_2P(O)(OH)_2$)).

### 3.5. Indole Detection

We qualitatively evaluated whether the strains could produce indole compounds, which are compounds related to the elongation and cell division stimulation. Our results indicated the production of indole compounds in isolates *P. fluorescens* E221, *Rahnella* sp. EM1, and *Serratia* sp. EM2.

### 3.6. Seed Germination

Two PGPB isolates remarkably affected the germination of maize seeds (Figure 3). The first seeds started to germinate (marked by radicle emergence) 72 h post-inoculation. The highest seed germination was recorded when the seeds were pretreated with isolates of *P. poae* EE12 and *P. fluorescens* E221, which increased seed germination at 72 h by 18.5% and 16.5%, respectively, with respect to the control. The length of the root on the 10th day ranged between 2.4 (SE = 0.076) cm for seeds inoculated with *B. Subtillis* 42 and 3.2 cm for seeds inoculated with *P. fluorescens* E221 and *P. poae* EE12 (SE = 0.086 and 0.091, respectively), which was statistically greater ($p < 0.0001$) than the control and the rest of the treatments (Figure 4). Germination is one of the most important and critical phases of the crop cycle, so a late or failed seedling emergence has a direct impact on yield [54]. In this context, *P. poae* EE12 and *P. florescens* E221 represent an alternative to a scenario with such an impact given their ability to significantly increase the percentage of germination and the length of the primary roots.

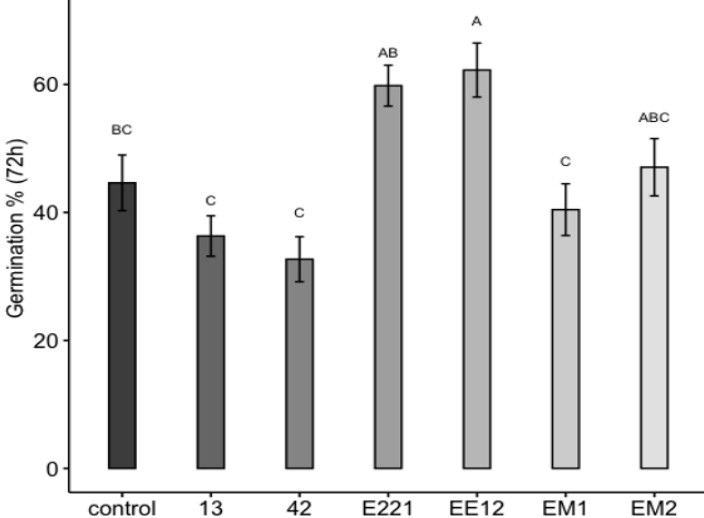

**Figure 3.** Effect of the inoculation of growth promoting bacteria 13, 42, E221, EE12, EM1, and EM2 on the in vitro germination of maize seeds (in a growth chamber with a 12/12-h light/dark cycle at 24 °C). Germination percentage of inoculated seeds at 72 h. Bars represent mean ± standard error and the different letters indicate significantly different values tested with one-way analyses of variance (ANOVA) and post-hoc Tukey-Kramer HSD test. Bars that do not share any letter are significantly different ($p = 2 \times 10^{-7}$), n = 200.

### 3.7. Early Growth Promotion Test in Seedbeds

A highly significant effect was observed, which was induced by the bacteria for the total area and the fresh and dry weight of the maize plants at 16 days in seedbed conditions. Treatments inoculated with strains *P. poae* EE12 and *Serratia* sp. EM2 had a significantly greater total area of the maize plants (Figure 5), unlike the rest of the treatments that did not show significant differences with respect to the control. In the same way, we observed that *P. poae* EE12 and *Serratia* sp. EM2 significantly increased the fresh weight of 16-day-old maize plants. Treatment with *Rahnella* sp. EM1 generated a higher dry weight compared to the control, unlike the rest of the treatments (Figure 6). The dry and fresh weight of the shoots and roots separately was also measured. We observed that EM2 significantly

increased the fresh weight of the root and shoot while *P. poae* EE12 only increased the fresh weight of the shoot (Figure 7A,B). Regarding the dry weight, we observed that the treatment with *Rahnella* sp. EM1 increased the dry weight of the root, while in the shoot we can see a tendency to increase the dry weight in treatments with *P. poae* EE12, *Rahnella* sp. EM1 and *Serratia* sp. EM2; however, there are no significant differences (Figure 7C,D). In contrast, *P. poae* E221 seemed to decrease both the area and the dry weight of the seedlings despite having shown positive results in promoting germination and having increased the length of the primary root under in vitro conditions.

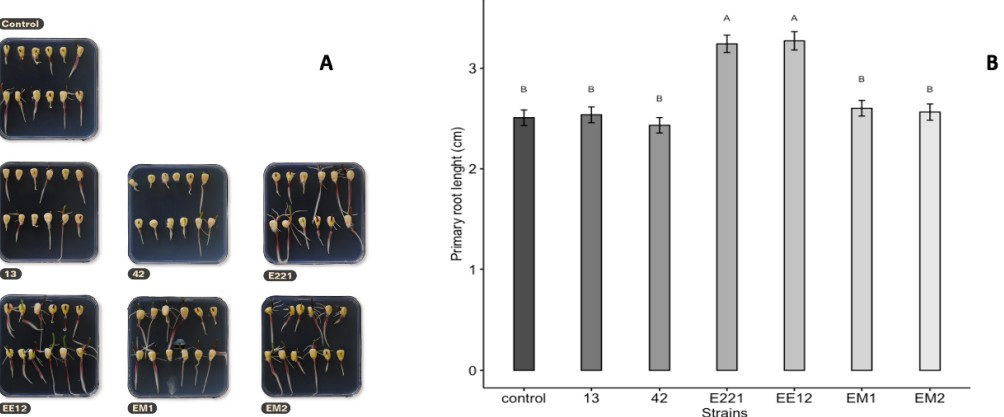

**Figure 4.** (**A**) Representative photographs of the effect of inoculation of growth promoting bacteria 13, 42, E221, EE12, EM1, and EM2 on primary root length on maize seeds in vitro at nine days. (**B**) Effect of inoculation of growth promoting bacteria on root length (cm) on maize seeds in vitro at 10 days (in a growth chamber with a 12/12-h light/dark cycle at 24 °C). Bars represent mean $\pm$ standard error and different letters indicate significantly different values tested with one-way analyses of variance (ANOVA) and post-hoc Tukey-Kramer HSD test. Bars that do not share any letter are significantly different ($p = 2 \times 10^{-16}$), n = 200.

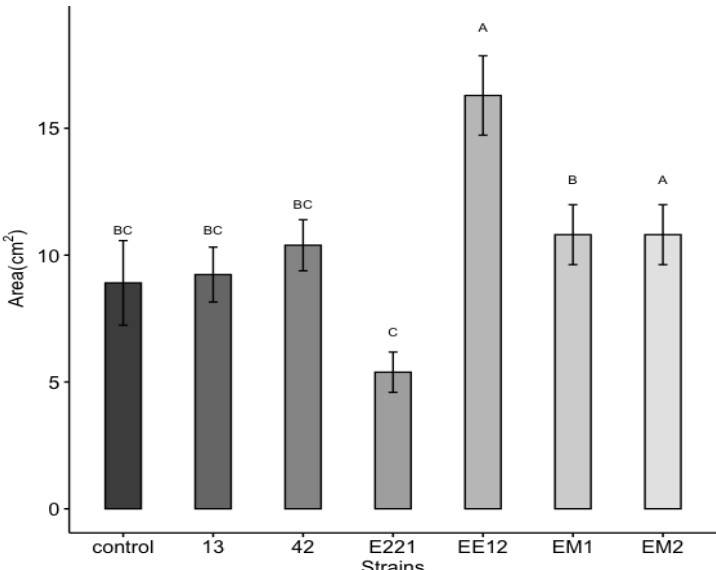

**Figure 5.** Effect of inoculation with plant growth promoting bacteria 13, 42, E221, EE12, EM1, and EM2 on the total plant area of maize seedlings in seedbeds at 16 days (in a growth chamber with a 12/12-h light/dark cycle at 24 °C). Bars represent mean $\pm$ standard error and different letters indicate significantly different values tested with one-way analyses of variance (ANOVA) and post-hoc Tukey-Kramer HSD test. Bars that do not share any letters are significantly different when the value of $p \leq 2.2 \times 10^{-6}$, n = 24.

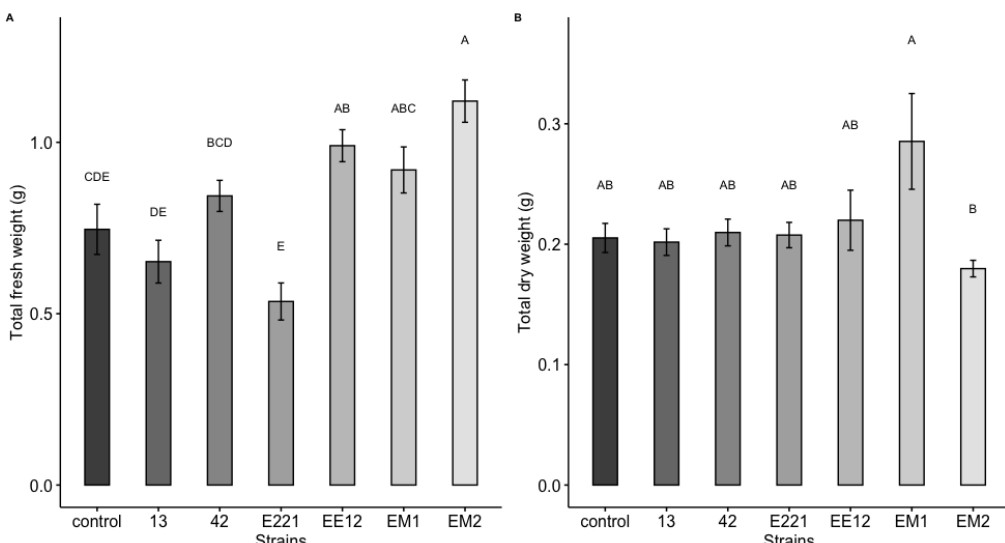

**Figure 6.** Effect of inoculation with plant growth promoting bacteria (13, 42, E221, EE12, EM1, and EM2) on (**A**) total fresh weight and (**B**) total dry weight of maize seedlings in seedbeds at 16 days (in a growth chamber with a 12/12-h light/dark cycle at 24 °C). In each subfigure, bars represent standard error, columns denoted by a different letter indicate significantly different values tested with one-way analyses of variance (ANOVA) and post-hoc Tukey-Kramer HSD test. Bars that do not share any letter are significantly different ($p < 0.05$), n = 24.

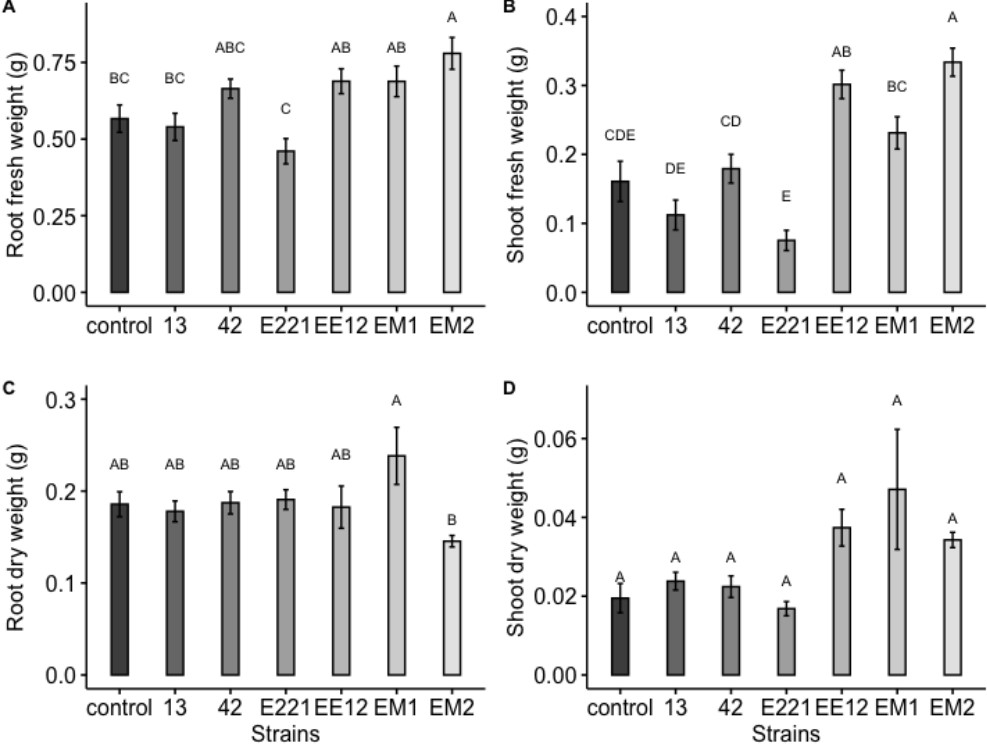

**Figure 7.** Effect of inoculation with plant growth promoting bacteria 13, 42, E221, EE12, EM1, and EM2 on early growth parameters of maize plants in seedbeds at 16 days (in a growth chamber with a 12/12-h light/dark cycle at 24 °C). (**A**) Root fresh weight, (**B**) shoot fresh weight, (**C**) root dry weight, and (**D**) shoot dry weight of maize seedlings. In each subfigure, the bars represent the standard error, the columns indicated with a different letter indicate significantly different values tested with one-way analyses of variance (ANOVA) and post-hoc Tukey-Kramer HSD test. Bars that do not share any letter are significantly different ($p < 0.05$), n = 24.

## 4. Discussion

In the present study, strains *B. subtillis* 42, *P. poae* EE12, and *B. velezensis* 13 demonstrated the ability to control the growth of *F. oxysporum* fungi. The *Bacillus* genus has been shown to effectively protect different crops, including maize, against different phytopathogenic fungi such as *Fusarium* sp., *Fusarium verticillioides,* and *Monilinia* spp. by producing chitinase, volatile, and other antifungal molecules [55–59]. *Pseudomonas* is another genus that has been widely studied for its biocontrol capacity, and it has shown antifungal activity against different phytopathogenic fungi such as *Verticillum* sp., *Rhizoctonia solani,* and *Fusarium* sp. [60–62] by producing a large group of metabolites such as 2,4-diacetylphloroglucinol, phenazine-1-carboxylic acid, and pyrrolnitrin [63]. Particularly, *P. poae* has been reported as a potential biocontrol agent: *P. poae* JSU-Y1 could inhibit the growth of *Penicillium expansum* on plate medium and apple fruit [64] and *P. poae* FL10F produces a cyclic lipopeptide belonging to the viscosin subfamily that possesses antagonistic activity against *Erwinia amylovora* [65]. Our results indicated that the isolated *B. subtillis* 42, *P. poae* EE12 and *B. velezensis* 13 would be promising bacterial resources to control phytopathogenic fungi in agricultural crops.

The plant growth-promoting activity of the studied strains includes not only the biocontrol capacity, but also additional properties such as the production of indole compounds related to auxin biosynthesis. Auxins are a group of plant growth regulators that stimulate cell division and elongation. Indole-3-acetic acid (IAA) can increase cell growth or proliferation [66]. However, it can also inhibit root growth in sugar beet and black currant [67], and these effects are concentration-dependent [68]. The PGPBs that produce auxins modify the root system by increasing the density of roots and changing their architecture [69,70]. This results in better nutrient exchange through well-developed roots [71]. Of the bacteria studied in this work, *P. fluorescens* E221, *Rahnella* sp. EM1 and *Serratia* sp. EM2 can produce indole compounds.

*Pseudomonas poae* EE12 and *P. fluorescens* E221 increased the seed germination percentage at 72 h with respect to the control, and the length of the primary root was significantly greater in the germinated seeds inoculated with *P. fluorescens* E221 and *P. poae* EE12 than in the control and in the rest of the treatments. Many studies have revealed that *P. fluorescens* strains had positive impacts on different growth parameters including germination percentage and seed vigor [72,73]. *Pseudomonas poae* has also been reported as a plant growth promoter species and IAA producer [74,75]. The ability of the strains to metabolize different P sources in vitro indicates that their inoculation into seeds could improve the utilization of soil P through two different processes: solubilization of soil inorganic phosphorus compounds (secondary minerals) [53] as the inorganic molecules tested ($FePO_4$, $AlPO_4$, and $Ca(H_2PO_4)_2H_2O$), and through the mineralization of organic molecules such as phosphonates (2AEP ($H_2NCH_2CH_2P(O)(OH)_2$)). The solubilization and mineralization processes are the basis of the increase of soil Pi availability for the biota. The mineralization of phosphonate (2AEP) is an enzyme-catalyzed process that occurs extracellularly and depends on the expression of the *phnX* gene in soil microorganisms. Solubilization occurs as a sub-product of enzyme-catalyzed processes [76] that depend on the expression of the *gcd* gene in soil microorganisms. However, *P. fluorescens* E221 seemed to decrease the area and the dry weight of the seedlings in the early growth promotion test in seedbeds, which could be attributed to the production of volatile organic compounds (VOCs) that under in vitro conditions could be promoting growth by triggering hormonal activities that are lost by decreasing the concentration of these compounds in an open system. Some of the VOCs with these capacities that have been reported are albuterol and 1,3-propanediol, 13-Tetradecadien-1-ol, 2-butanone, and 2-Methyl-n-1-tridecene [77,78]. To find out if this is the cause of this response, tests can be performed by increasing the concentration of bacteria.

*Pseudomonas poae* EE12 was able to increase in seedbed conditions the total area and shoot fresh weight of maize plants. These results are consistent with similar studies where *P. poae* EE12 enhanced various parameters that evaluate plant growth [79,80]. *Serratia* sp. EM2 significantly increased the total fresh weight. *Serratia* sp. SY5 has shown capacity to

produce indole acetic acid and the synthesis of siderophores and root growth promotion of *Zea mays* seedlings [81]. In other studies, on *Serratia*, various mechanisms have been reported, such as the solubilization of phosphorus, the production of siderophores and IAA, among others [82–85]. In *Rahnella*, multiple traits that promote plant growth such as the solubilization of organic and inorganic phosphate, 1-aminocyclopropane-1-carboxylate-deaminase activity, the generation of ammonia, and production of siderophores, have been reported [86–89].

The six strains analyzed herein represent an alternative to reduce the use of agrochemicals and the damage they cause to human health and the environment. However, further studies are needed, e.g., to study the use of combinations of strains or even the evaluation of a consortium formed by the six strains, since the PGPB can be used alone or in consortia and always with an elevated positive impact on agricultural production [90].

**5. Conclusions**

In summary, *B. velezensis* 13 and *B. subtillis* 42 have the greatest capacity to biocontrol the phytopathogenic fungi *B. cinerea* and *F. oxysporum*, and the rest of the strains have capacities that make them suitable as plant growth promoters. *P. poae* EE12 and *P. fluorescens* E221 significantly promote germination and the length of primary root. *Rahnella* sp. EM1 and *Serratia* sp. EM2 produce indole compounds related to auxin synthesis and increase the fresh and dry total weight of the maize seedlings. It would be interesting to carry out experiments with the entire consortia, since the six bacteria together are a promising biocontrol and growth promotion agent.

In conclusion, the six PGPB isolated from the soil and litter of Mexican oak forests have the potential to play an important role in crop production, in addition to the fact that due to their origin, these bacteria can be tested as recovery, reforestation, and biocontrol agents in forestry [91–94]. Soil microorganisms from Mexican oak forests represent a genetic source for the sustainable management and conservation of soils.

**Author Contributions:** Conceptualization, R.H.-L., A.G.-R. and Y.T.-T.; Formal analysis, R.H.-L.; Funding acquisition, Y.T.-T.; Resources, Y.T.-T.; Writing—original draft, R.H.-L.; Writing—review & editing, A.G.-R. and Y.T.-T. All authors have read and agreed to the published version of the manuscript.

**Funding:** This research was funded by the "Application of ecological knowledge to favor the sustainability of avocado cultivation in Michoacán state: soil, hydrological and biotic interaction aspects, PFCTI/ICTI/2019/A/315", "Collaborative network for the teaching of biogeochemistry in Mexico through a virtual laboratory based on case studies (PAPIME-UNAM-PE206922)", and "The APC paid by authors".

**Data Availability Statement:** DNA sequences are available at https://www.ncbi.nlm.nih.gov (access numbers: OK324128, OK324129, OK324130, OK324131, OK324132, and OK324133). https://www.ncbi.nlm.nih.gov/guide/genes-expression/.

**Acknowledgments:** The authors are thankful for the CONACyT postdoctoral stay program (CVU: 268952) and the PAPIME-UNAM grant (PE206922).

**Conflicts of Interest:** The funders had no role in the design of the study; in the collection, analyses, or interpretation of data; in the writing of the manuscript, or in the decision to publish the results.

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
