# Peer review of "Phosphorus Recycling, Biocontrol, and Growth Promotion Capabilities of Soil Bacterial Isolates from Mexican Oak Forests: An Alternative to Reduce the Use of Agrochemicals in Maize Cultivation"

_2673-8007, doi:10.3390/applmicrobiol2040074_

Round 1
Reviewer 1 Report
Line 14, “P sources” should be “phosphorus sources”.
Line 187, “de” should be “the”?
In Figure 1: The representative plates are shown. Could the authors include the strain information for each plate? It would be better to show the control plates in parallel with representative plates.
In Table 1: (1) Should be 16S rRNA gene; (2) Could the authors explain “-” for Siderphores production? Does it mean “undetectable” or “no production”?
Line 264, “completely” should be changed to another word, e.g., dramatically
Line 268, “cinereal” should be “cinerea”.
For table 1: Could the authors explain why the “indole production” analysis was shown in this table? Is the indole production associated with growth inhibition of fugus?
In Figure 3-7, what is the meaning of “BC, C, ABC..” above the plot? Please further describe the statistical analysis, the meaning of A, B, C, p value in the figure legend.
Reviewer 2 Report
The work is interesting, but the experimental part needs to be refined.

Round 2
Reviewer 2 Report
The paper can be accepted in this form.
Author Response
Thank you